# Development and initial validation of the German version of the Exergame Enjoyment Questionnaire (EEQ-G)

Patrick Manser[1]*, Simone Huber[1,2], Julia Seinsche[1], Eling D. de Bruin[1,3,4], Eleftheria Giannouli[1,5]

1 Department of Health Sciences and Technology, Motor Control and Learning Group–Institute of Human Movement Sciences and Sport, ETH Zurich, Zurich, Switzerland, 2 Physiotherapy and Occupational Therapy Research Centre, Directorate of Research and Education, University Hospital Zurich, Zurich, Switzerland, 3 Division of Physiotherapy, Department of Neurobiology, Care Sciences and Society, Karolinska Institute, Stockholm, Sweden, 4 Department of Health, OST–Eastern Swiss University of Applied Sciences, St. Gallen, Switzerland, 5 Division of Sports and Exercise Medicine, Department of Sport, Exercise and Health, University of Basel, Basel, Switzerland

* patrick.manser@hest.ethz.ch

## Abstract

### Background

Analyzing and adjusting training programs to increase exercise enjoyment is crucial to achieve long-term adherence and thus also maximize health benefits. The Exergame Enjoyment Questionnaire (EEQ) is the first questionnaire specifically developed to monitor exergame enjoyment. To be used in German speaking countries, the EEQ must be translated, cross-culturally adapted, and tested on its psychometric properties.

### Objectives

The aim of this study was to develop (i.e., translate and cross-culturally adapt) the German Version of the EEQ (EEQ-G) and investigate its psychometric properties.

### Methods

Psychometric properties of the EEQ-G were tested using a cross-sectional study design. Each participant performed two consecutive exergame sessions (i.e., 'preferred' and 'unpreferred' condition) in randomized order and rated the EEQ-G as well as reference questionnaires. Internal consistency of the EEQ-G was assessed by calculating Cronbach's α. Construct validity was assessed by calculating Spearman's rank correlation coefficients ($r_s$) between the scores of the EEQ-G and reference questionnaires. Responsiveness was analyzed by performing a Wilcoxon signed-rank test between the median EEQ-G scores of the two conditions.

### Results

Fourty-three healthy older adults (HOA; mean age = 69.4 ± 4.9 years; 53.5% females) were included. Cronbach's α of the EEQ-G was 0.80. The $r_s$ values between the EEQ-G and

**Data Availability Statement:** The datasets generated and/or analyzed during the current study are available in the Zenodo repository, https://doi.org/10.5281/zenodo.7373180.

**Funding:** The authors received no specific funding for this work.

**Competing interests:** The authors have declared that no competing interests exist.

**Abbreviations:** BREQ, Behavioral Regulation in Exercise Questionnaire; CI$_{95\%}$, 95% confidence interval; EEQ, Exergame Enjoyment Questionnaire; EEQ-G, German Version of the Exergame Enjoyment Questionnaire; GEeQ, Game Engagement Questionnaire; GExQ, Game Experience Questionnaire; H$_0$, Null Hypothesis; H$_A$, Alternative Hypothesis; HOA, Healthy Older Adults; IEQ, Immersive Experience Questionnaire; IPAQ-SF, International Physical Activity Questionnaire—Short Form; IQR, Interquartile Range; MET, Metabolic Equivalent of Tasks; PACES, Physical Activity Enjoyment Scale; r$_s$, Spearman's rank correlation coefficients; SD, Standard Deviation.

reference questionnaire scores for intrinsic motivation, game enjoyment, physical activity enjoyment, and external motivation were 0.198 (p = 0.101), 0.684 (p < 0.001), 0.277 (p = 0.036), and 0.186 (p = 0.233), respectively. The EEQ-G was rated higher in the 'preferred' than the 'unpreferred' condition (p < 0.001, r = 0.756).

## Conclusion

The EEQ-G has high internal consistency and is responsive to changes in exergame enjoyment. The highly skewed data with ceiling effects in some of the reference questionnaires deem the construct validity of the EEQ-G to be inconclusive and thus in need of further evaluation.

## Introduction

### Background

Exercise enjoyment has been described as *"an optimal psychological state (i.e., flow) that leads to performing an activity primarily for its own sake and is associated with positive feeling state"* [1]. People are more likely to enjoy exercising if it induces feelings of competence, relatedness, and autonomy [2, 3]. In this context, exercise enjoyment is intimately linked with intrinsic motivation and positive affect [1], important factors for promoting positive behavioral changes [4] (e.g., adherence to exercise) in healthy adults [5–9], HOA [3, 7, 10, 11], and in patients with chronic diseases [3, 12]. For example, exercise enjoyment was predictive for future physical activity participation in low-active adults [13], and more autonomous forms of motivation predicted physical training frequency, intensity, and duration in a large cohort of regular exercisers [5]. Importantly, perceived enjoyment and training adherence have also been shown to moderate the efficacy of training interventions in different populations [14–16]. Therefore, analyzing and adjusting training programs to increase exercise enjoyment (and thereby adherence) is important to achieve continuous health benefits [17]. To do this, it is important to have valid and reliable instruments to assess exercise enjoyment. In physical activity and/or training settings, exercise enjoyment is commonly measured using the Physical Activity Enjoyment Scale (PACES), originally developed and validated by Kendzierski and DeCarlo in 1991 [18].

Technological innovations provide new options to engage individuals in physical activity or training programs. They can–for example—be applied in form of an 'exergame'–"[. . .] defined as technology-driven physical activities, such as video game play, that requires participants to be physically active or exercise in order to play the game" [19]. Exergames can help to overcome many perceived barriers to exercise because they are engaging [20, 21], provide immediate performance feedback (e.g., visual, auditory, tactile) that enriches the training experience [20], can be individually tailored and progressed in real-time [22–24], and allow individuals to engage in their training program at home [20]. In turn, older adults are then more likely to engage for example in falls prevention exercise programs [25]. Indeed, exergaming has shown to be an enjoyable form of training [21].

Depending on the population and the type of exercise(s), different factors may determine how training enjoyment can be achieved and/or sustained. In this regard, the needs of the specific users should be taken into account when designing such technologies [20] in order to make the exergame experience (even more) enjoyable. As an example, using exergames can be difficult for older adults, especially for those who have little or no experience with technology

or in case the system lacks clear instructions, is fast-paced, or presents too much graphical information [20]. This will most likely have a negative effect on exergame enjoyment and adherence. Despite these limitations, technology-based (including exergame-based) training programs typically have higher adherence rates as compared to conventional training in older adults, which has been explained by the high reported levels of enjoyment when using these programs [20]. However, most studies investigating exergame enjoyment have applied semi-structured interviews [20, 21] or have used physical activity enjoyment scales/questionnaires such as the PACES [26–35]. These assessments aim to assess rather conventional types of training and are not specifically adapted for technology-based training. To ensure the credibility and comparability of investigations on exergame enjoyment, an exergame-specific instrument that is valid, reliable, and sensitive to changes in exergame enjoyment is needed.

Fitzgerald and colleagues developed and validated the EEQ [36]. The EEQ is the first questionnaire specifically developed to assess and monitor exergame enjoyment. The EEQ combines elements of well-known and widely used questionnaires to assess physical activity enjoyment (i.e., PACES [18]) and gameplay (i.e., Game Engagement Questionnaire (GEeQ) [37], Game Experience Questionnaire (GExQ) [38], and Immersive Experience Questionnaire (IEQ) [39]) and adds new elements that are specifically relevant for exergaming. To the best of our knowledge, there is no validated German version of the EEQ available. To be used in German speaking countries, the EEQ must be translated and cross-culturally adapted. Additionally, it must be tested on its psychometric properties to ensure the comparability of responses across populations [40]. The translated and cross-culturally adapted questionnaire should retain internal consistency, construct validity and responsiveness [40]. Internal consistency is the degree of interrelationship / homogeneity among items to measure the same construct [41]. Construct validity describes the extent to which a questionnaire accurately assesses the construct it is supposed to measure, including convergent (i.e., constructs that are expected to be related are, in fact, related) and discriminant (i.e., constructs that should have no relationship do, in fact, not have any relationship) validity [41]. Finally, responsiveness stands for the extent to which an instrument/questionnaire can detect changes in the construct being measured over time [42].

### Objectives

The aim of this study was to develop (i.e., translate and cross-culturally adapt) the German Version of the EEQ (EEQ-G) and to investigate its psychometric properties in terms of internal consistency, construct validity, and responsiveness in HOA.

Regarding construct validity, it was expected that exergame enjoyment positively relates to intrinsic motivation, physical activity enjoyment, and enjoyment of gameplay (convergent validity) but is not related to external motivation (discriminant validity). Therefore, the following alternative hypotheses were defined: For convergent validity, it was hypothesized that, in HOA, there is a significant large positive correlation between the EEQ-G rating and ($H_{A,1}$:) intrinsic motivation, ($H_{A,2}$:) enjoyment of gameplay, ($H_{A,3}$:) physical activity enjoyment. For discriminant validity, it was hypothesized that, in HOA, there is no correlation between the EEQ-G rating and external motivation ($H_{A,4}$).

## Materials and methods

### Trial design and study setting

First, the original English EEQ was translated and cross-culturally adapted to German according to the *"guidelines for the process of cross-cultural adaptation of self-report measures"* [40]. We included two bilingual translators whose mother tongue is German for the forward

translation. Two other bilingual translators with the source language (English) as their mother tongue did the back translation. All translators fulfilled the criteria defined in [40]. In line with these guidelines, we obtained approval of the original developer of the questionnaire to develop a translated version of the questionnaire, and the original developer approved the translated version.

Second, a cross-sectional study with HOA was conducted. After recruitment and providing written informed consent, participants were screened for eligibility (see section 'Participants and Recruitment') and the measurements were conducted (see section 'Measurement Procedures'). All study procedures took place at ETH Hönggerberg in one study session per participant and were led by one investigator of the research team trained in the application of the measurement techniques and protocols.

## Participants and recruitment

HOA were recruited by contacting (by mail or phone call) suitable individuals from a participant pool consisting of participants who expressed interest in participating in future studies of our research group. Subsequently, an appointment for study participation was scheduled, where all interested persons first provided written informed consent, were then screened on eligibility, and finally went through all the measurement procedures (see Fig 1 and section 'Measurement Procedures'). All eligibility criteria are detailed in Table 1.

## Measurement procedures

At first, all participants were familiarized with the exergame training system 'Senso' (Dividat AG, Schindellegi, Switzerland) and the five exergames used in this study (i.e., 'Simple', 'Targets', 'Habitats', 'Tetris', 'Simon'; video demonstrations of the games see [43]). Subsequently, each participant was asked to rank-order these five exergames according to their preferences. Based on this ranking, each participant performed two consecutive exergame sessions relating to two conditions that differed in content according to the participants' preferences. In the 'preferred' condition, participants played the two games they preferred most, including auditory feedback, for 3 minutes each. In the 'unpreferred' condition, participants played their least preferred game twice for 3 minutes. In addition to having less preferred games and less content variance compared to the 'preferred' condition, auditory feedback was removed from

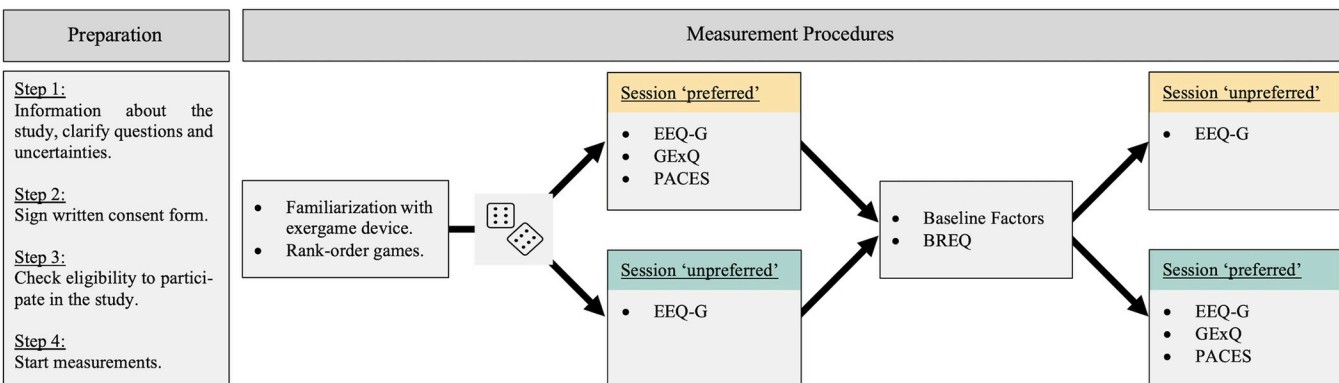

**Fig 1. Overview of the study procedures.** The cubes are used to illustrate the randomization process (variable block randomization (i.e., block sizes = 4, 6, 8) with a 1: 1 allocation ratio and stratified by sex; Colour coding: orange = group 1 (starts with 'preferred' condition, green = group 2 (starts with 'unpreferred' condition); Abbreviations: BREQ, Behavioral Regulation in Exercise Questionnaire; EEQ, Exergame Enjoyment Questionnaire; GExQ, Game Experience Questionnaire; IQR, Interquartile Range; PACES, Physical Activity Enjoyment Scale.

**Table 1. Description of all eligibility criteria.**

| Inclusion criteria | Exclusion criteria |
|---|---|
| Participants fulfilling all the following inclusion criteria were eligible: | The presence of any of the following criteria led to exclusion: |
| • healthy (based on self-report) older adults (≥ 60 years)<br>• ability to stand for at least 10 min without assistance<br>• German speaking | • mobility impairments (i.e., gait, balance) that prevent from study participation<br>• presence of neurological disorders (i.e., epilepsy, stroke, multiple sclerosis, Parkinson's disease, brain tumors, or traumatic disorders of the nervous system) |

the game in this condition, considering that these factors relate to decreased exergame enjoyment [20, 44, 45]. In both conditions, the task demands were individually adapted according to the participant's performance using the internal progression algorithm of the exergame system. The two conditions were completed in randomized order. Randomization was performed using a validated variable block randomization model (block sizes = 4, 6, 8; stratified by sex) implemented in the data management system Castor EDC (Ciwit BV, Amsterdam, The Netherlands) [46]. Participants were informed that two different exergame sessions would be conducted consecutively but were not provided with any information on how these two sessions differed in content. Participants rated their exergame enjoyment immediately after completing each condition. Additionally, participants filled out reference questionnaires to rate their physical activity enjoyment and gameplay after the 'preferred' condition. In between the two sessions, baseline factors of the participants were collected, and the participants filled out a questionnaire about their exercise motivation. Fig 1 provides an overview of the study session.

## Outcomes

**Baseline factors.** Baseline factors were collected through demographic data including age, sex, years of education, physical activity behavior (i.e., measured with the German version of the International Physical Activity Questionnaire—Short Form (IPAQ-SF) [47, 48] and analyzed according to guidelines for data processing and analysis of the IPAQ-SF [49]), analogue (including activities such as board and card games, crosswords, or Sudoku) and digital (including activities such computer games, and game consoles) gaming activity behavior measured as total time spent on analogue/digital games per week [min/week], and exergame training experience with the 'Senso' (yes / no, if yes: once, 2–5 times, > 5 times).

**Exergame enjoyment.** Exergame enjoyment was assessed with the EEQ-G (supplementary file 1; description see section 'Introduction'). The questionnaire consists of 20 statements rated on a five-point Likert scale (i.e., 'strongly disagree', 'disagree', 'neutral', 'agree', and 'strongly agree') corresponding to four categories of questions: (1) immersion, (2) intrinsically rewarding activity, (3) control, and (4) exercise. The EEQ-G was scored by adding up the points for each statement, as described in the scoring guidelines (see supplementary file 1). This results in a minimum possible score of 20 and a maximum possible score of 100. A higher score reflects greater exergame enjoyment.

**Training motivation.** Training motivation was assessed with the German translation [50] of the revised [51] Behavioral Regulation in Exercise Questionnaire (BREQ) [52], a widely used, valid and reliable measure of training motivation [7, 51–54] with acceptable internal consistency (Cronbach's α between 0.73 and 0.91 [51, 53]. The BREQ uses a Likert scale with items for motivational subscales along the Self-Determination Continuum [52, 55]. The German translation of the BREQ has been used to assess change in training motivation over time in a randomized controlled trial (RCT) investigating a lifestyle-integrated functional exercise intervention in older adults (n = 294). In this study, the German translation of the BREQ had a

high internal consistency (Cronbach's α of 0.84–0.87) [56]. Additionally, it has been used in a pilot RCT investigating a computerized cognitive training approach aiming to increase physical activity in healthy individuals [57] and a cross-sectional study testing an extension of the health action process approach by including intrinsic motivation in older adults [58].

The BREQ consists of 24 items assessing behavioral regulation in exercise contexts rated on a 5-point Likert scale ranging from 0 = 'not true for me' to 4 = 'very true for me'. Mean values for the ratings of each motivational subscale (i.e., external, integrated, introjected, identified, and intrinsic motivation as well as amotivation) were calculated to assess qualitatively different forms of behavioral regulation [59]. Additionally, the Self Determination Index (SDI) was calculated as described in [59], a valid and reliable composite score to simplify individual representation along the continuum of self-determination. SDI ranges between—24 and + 24, where higher positive scores represent a higher degree of self-determined motivation [59].

**Enjoyment of gameplay.** The GExQ assesses seven categories of subjective game experience (i.e., competence, immersion, flow, tension, challenge, negative affect, and positive affect) with acceptable internal consistency (Cronbach's α between 0.69 and 0.91 [38, 60], and has been applied in studies evaluating exergaming experience in older adults [61, 62]. The German GExQ core module was used in this study, which includes 12 items rated on a 5-point Likert scale (0 = 'not at all' to 4 = 'absolutely'). It was analyzed according to published scoring guidelines by calculating the average overall score [63].

**Physical activity enjoyment.** The degree of physical activity enjoyment during the exergame session was assessed using the PACES [18]. The PACES has already been implemented for measuring exergame enjoyment in HOA in multiple studies [29–32, 35]. The German [64] 16-item version [65] of the PACES was used, which is valid and reliable in children, adolescents and adults [64–66], with high internal consistency (Cronbach's α between 0.89 to 0.94) [64, 66]. The German PACES has been used in numerous studies, as for example in RCTs investigating the effects of different types of motor-cognitive training on cognitive performance in older adults [67, 68], or to assess psychological and physiological responses to exergaming [69].

Items of the German PACES are rated on a 5-point Likert scale (1 = 'do not agree at all' to 5 = 'fully agree' for positive items (i.e., items 1, 4, 6, 8, 9, 10, 11, 14, 15), and 5 = 'do not agree at all' to 1 = 'fully agree' for all remaining negative items), and were analyzed by summing up all items to a total score for each participant [64].

**Data management.** All involved study investigators were thoroughly trained for all study procedures according to Guidelines of Good Clinical Practice (GCP) and in line with detailed working instructions. The principal investigator was in charge for methodological standards and quality of data collection using data management system Castor EDC (Ciwit BV, Amsterdam, The Netherlands) [46]. Range checks for data values were pre-programmed for data entry in eCRFs. All data entries were cross-checked by a second study investigator prior to export for analysis. To minimize bias during assessment of all outcome measures, detailed working instructions were prepared that include standardized measurement procedures and standardized instructions of participants for all measurements.

**Analytical methods.** Statistical analysis was executed using R Version R 3.6.2 GUI 1.70 El Capitan build (7735) (© The R Foundation) in line with RStudio Version 2022.07.1 (RStudio, Inc.). Questionnaire scores were regarded as ordinal data. Data was reported as mean ± standard deviation for continuous parametric data and median (interquartile range) for continuous non-parametric data.

First, descriptive statistics were computed for all outcome variables. Normality distribution of data was checked using the Shapiro-Wilk test. The level of significance was set to $p \leq 0.05$ (two-sided). Statistical analysis was done by PM after data collection was completed.

**Internal consistency.** Cronbach's $\alpha$ was calculated to investigate the internal consistency of the EEQ-G [70]. The degree of consistency was interpreted according to the categorization for Cronbach's $\alpha$ defined in [41]. Cronbach's $\alpha \geq 0.70$ was set as the criterion for "adequate" internal consistency [41].

**Construct validity.** Bivariate correlation analyses between the EEQ-G score of the 'preferred' condition and each of the corresponding references for the defined hypotheses: (1) the sub-score 'intrinsic regulation' of the BREQ, (2) the total PACES score, (3) the total GExQ score, and (4) the sub-score 'external regulation' of the BREQ. Spearman's rank correlation coefficients ($r_s$) were calculated and interpreted to be small ($0.1 \leq |r_s| < 0.3$), medium ($0.3 \leq |r_s| < 0.5$) or large ($|r_s| \geq 0.5$) [70, 71].

For convergent validity, the alternative hypotheses were considered confirmed in case of: (a) a significant ($p \leq 0.05$, one-tailed) positive correlation between the EEQ-G score and the corresponding reference (see hypotheses 1–3); and (b) a validity coefficient of $|r_s| \geq 0.5$. For discriminant validity, the alternative hypothesis was considered verified in case of no significant ($p \geq 0.05$, two-tailed) correlation between the EEQ-G score and the corresponding reference (see hypothesis 4).

**Responsiveness.** For the context of this study, we defined that responsiveness refers to the ability of the EEQ to detect changes in exergame enjoyment in relation to differing (preferred versus unpreferred) consecutive exergaming conditions. On this basis, responsiveness was analyzed performing a Wilcoxon signed-rank test evaluating whether there is a difference in median EEQ-G scores between the two conditions ('preferred' vs. 'unpreferred'). To discover whether the effects were substantive, effect sizes r were calculated [70, 72] and interpreted to be small ($0.1 \leq r < 0.3$), medium ($0.3 \leq r < 0.5$) or large ($r > 0.5$) [71]. Responsiveness of the EEQ-G was considered as given in case there was a significant ($p \leq 0.05$) difference in median EEQ-G scores between the two conditions ('preferred' vs. 'unpreferred') with at least a medium effect size ($r \geq 0.4$).

**Sample size justification.** For internal consistency, a sample size of n = 14 is required to achieve sufficient power ($1 - \beta \geq 0.80$) when considering the criterion of Cronbach's $\alpha \geq 0.70$ for adequate internal consistency [41], the number of items in the questionnaire of k = 20, and a level of significance of $\alpha = 0.05$ [73]. For construct validity, a-priori sample size estimation was calculated using G*Power 3 software [74, 75]. Given a required validity coefficient of $|r_s| \geq 0.5$ and the level of significance of $\alpha = 0.05$, a sample size of n = 29 is required to achieve sufficient power ($\beta > 0.80$). For responsiveness, a sample size of n = 42 is required to achieve sufficient power ($\beta > 0.80$) to fulfill the criteria of at least a medium effect size ($r \geq 0.4$) at a level of significance of $\alpha = 0.05$ according to the standard normal, exact variance method [76]. Taken together, the evaluation of responsiveness requires the largest sample size to reach sufficient statistical power. To ensure an adequate number of participants in the study, a safety margin for missing data due to drop-outs or technical problems of 10% was chosen. Therefore, we aimed to recruit 42–46 participants.

## Results

### Recruitment and participant flow

A summary of the participant flow through the study is illustrated in Fig 2. Recruitment was stopped when complete data of the planned minimum sample size of 42 participants was available.

### Baseline data and descriptive statistics

The baseline factors of the study participants are summarized in Table 2. The game most often ranked as "most preferred" was 'Habitats' (n = 19, 44% of participants), followed by 'Simple'

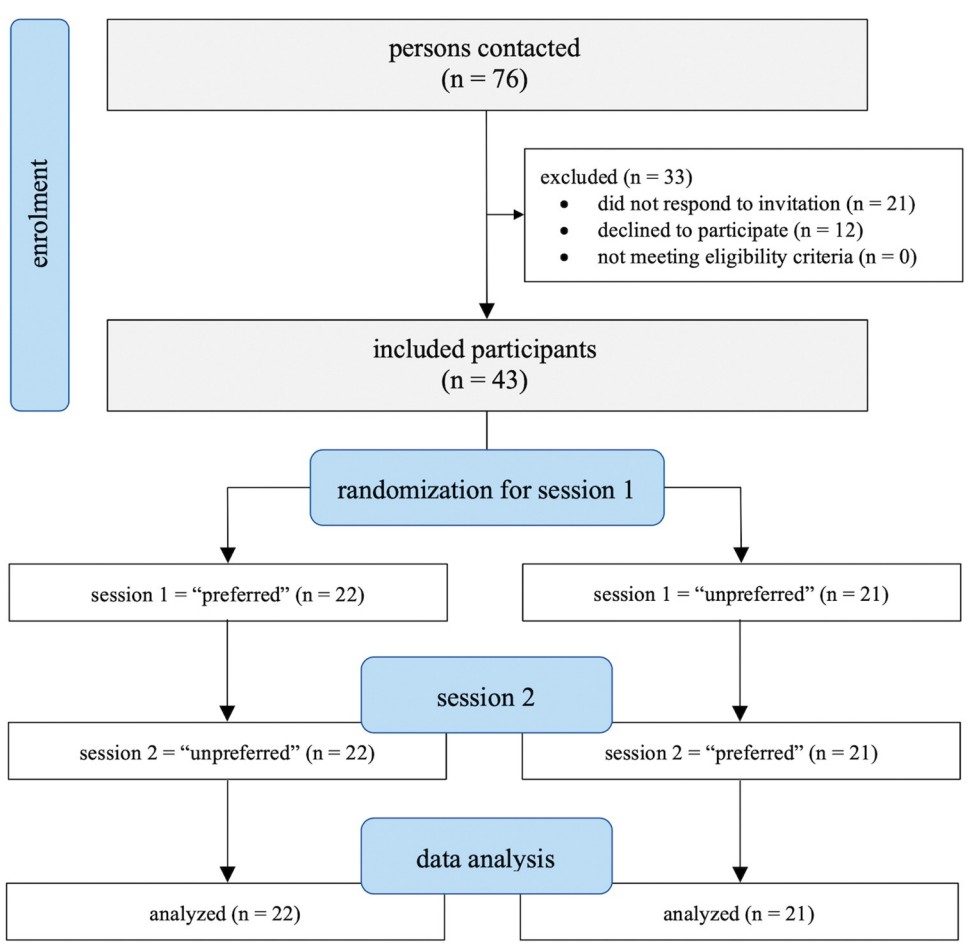

**Fig 2. Summary of the participant flow throughout the study.**

(n = 9, 21% of participants), 'Targets' (n = 6, 14% of participants), 'Simon' (n = 5, 12% of participants), and 'Tetris' (n = 4, 9% of participants). The game most often ranked as "least preferred" was 'Tetris' (n = 25, 58% of participants), followed by 'Simon' (n = 10, 23% of participants), and 'Simple' (n = 8, 19% of participants).

**Table 2. Demographic characteristics of the study population.** Data is reported as mean ± standard deviation for continuous parametric data and median (interquartile range) for continuous non-parametric data.

| | Total Sample (n = 43) |
|---|---|
| Age [years] | 69.4 ± 4.9 |
| Education [years] | 15.9 ± 4.2 |
| Physical Activity [METmin · week$^{-1}$] | 6,132 (5,792) |
| Analogue Gameplay [min · week$^{-1}$] | 30 (92) |
| Digital Gameplay [min · week$^{-1}$] | 0 (40) |
| Previous experience with the specific exergame device before study participation: | |
| none | n = 7 (16% of participants) |
| once | n = 4 (9% of participants) |
| two to five times | n = 28 (65% of participants) |
| more than five times | n = 4 (9% of participants) |

Abbreviations: SD, Standard Deviation; MET, Metabolic Equivalent of Tasks

**Table 3. Descriptive statistics of the German version of the Exergame Enjoyment Questionnaire (EEQ-G) in each condition and the reference questionnaires.**

| Questionnaire | Median | IQR | Min | Max |
|---|---|---|---|---|
| EEQ-G | | | | |
| Total score; condition = 'preferred' (scores between 20 and 100) | 82.0 | 10.5 | 57 | 93 |
| Total score; condition = 'unpreferred' (scores between 20 and 100) | 69.0 | 18.0 | 40 | 88 |
| BREQ | | | | |
| Self Determination Index (scores between -24 and +24) | 17.0 | 5.5 | -12.00 | 22.50 |
| Subscore 'intrinsic motivation' (scores between 0 and 4) | 3.5 | 0.9 | 0.00 | 4.00 |
| Subscore 'integrated motivation' (scores between 0 and 4) | 3.0 | 1.4 | 0.00 | 4.00 |
| Subscore 'identified motivation' (scores between 0 and 4) | 3.5 | 0.9 | 0.50 | 4.00 |
| Subscore 'inrojected motivation' (scores between 0 and 4) | 1.5 | 1.3 | 0.25 | 4.00 |
| Subscore 'external motivation' (scores between 0 and 4) | 0.0 | 0.3 | 0.00 | 2.00 |
| Subscore 'amotivation' (scores between 0 and 4) | 0.0 | 0.0 | 0.00 | 3.00 |
| GExQ | | | | |
| Total score (scores between 0 and 48) | 29.0 | 5.5 | 13 | 35 |
| PACES | | | | |
| Total score (scores between 0 and 80) | 78.0 | 4.5 | 68.0 | 80.0 |

Abbreviations: BREQ, Behavioral Regulation in Exercise Questionnaire; GExQ, Game Experience Questionnaire; IQR, Interquartile Range; PACES, Physical Activity Enjoyment Scale

The descriptive statistics of the EEQ-G in each condition and the reference questionnaires (i.e., BREQ, PACES, and GExQ) are summarized in Table 3.

## Internal consistency

Cronbach's α of the EEQ-G questionnaire was 0.80 ($CI_{95\%}$ [0.72 to 0.88]).

## Construct validity

The $r_s$ values between the EEQ-G score and the scores of the corresponding reference questionnaires for each hypothesis are summarized in Table 4. Game experience and enjoyment of

**Table 4. Spearman's rank correlation coefficients ($r_s$) and p-values of the bivariate correlation analyses between the German version of the Exergame Enjoyment Questionnaire and the reference questionnaires.**

| Questionnaire | EEQ-G | |
|---|---|---|
| | $r_s$ | p-value |
| Convergent Validity: | | |
| BREQ–subscore 'intrinsic motivation' | 0.198 | 0.101 |
| GExQ–total score | 0.684 | < 0.001** |
| PACES–total score | 0.277 | 0.036* |
| Discriminant Validity: | | |
| BREQ–subscore 'external motivation' | 0.186 | 0.233 |

\* = significant at p < 0.05

\*\* = significant at p < 0.01

Abbreviations: BREQ, Behavioral Regulation in Exercise Questionnaire; GExQ, Game Experience Questionnaire; IQR, Interquartile Range; PACES, Physical Activity Enjoyment Scale -German version; rs, spearman's rank correlation coefficients

# Responsiveness of the EEQ-G

**Fig 3. Boxplot of the German version of the EEQ (EEQ-G) scores in the 'preferred' and the 'unpreferred' condition.**

physical activity were significantly correlated with exergame enjoyment, with a large and small correlation coefficient, respectively.

## Responsiveness

Exergame enjoyment was rated significantly higher in the 'preferred' (82.0 (10.5)) than the 'unpreferred' (69.0 (18.0)) condition ($p < 0.001$; see Fig 3), with a large effect size (r = 0.756).

## Discussion

The aim of this study was to develop (i.e., translate and cross-culturally adapt) the German Version of the EEQ (EEQ-G) and to investigate its psychometric properties in HOA. The results reveal (1) that the EEQ-G has high internal consistency; (2) inconclusive results for the construct validity of the EEQ-G; and (3) that the EEQ rating is responsive to changes in exergame enjoyment in relation to differing consecutive exergaming conditions.

### Internal consistency

Both, the mean and the $CI_{95\%}$ of Cronbach's α were above the criterion for adequate internal consistency, suggesting high internal consistency of the EEQ-G in the present population and setting [41]. To the best of our knowledge, no internal consistency analysis has been published for the original English EEQ. Therefore, our results cannot be compared with values from the original questionnaire. Nonetheless, our results are consistent with findings in the GExQ [60] and the PACES [64, 64, 77], from which several items were adopted for the development of the EEQ [36]. In particular, the dimensions 'immersion', 'flow' and 'competence' of the GExQ, which correspond with the dimensions of the EEQ-G, have shown comparably high internal consistency as the total score of the EEQ-G (immersion: α = 0.85, $CI_{95\%}$ [0.83, 0.87]; flow: α =

0.86, CI$_{95\%}$ [0.84, 0.88]; competence: $\alpha$ = 0.85, CI$_{95\%}$ [0.83, 0.86]) [60]. Moreover, the PACES has shown high internal consistency in various populations and languages (including German; i.e., Cronbach's $\alpha$ of 0.95 middle-aged to older adults with functional limitations [77], 0.94 in German young adults [66], and 0.89 to 0.92 in German adolescents [64], respectively).

## Construct validity

For convergent validity, we formulated three alternative hypotheses (see section 'Introduction–Objectives'). We expected large correlations between exergame enjoyment and intrinsic motivation (H$_{A,1}$), because intrinsic motivation refers to behavior that is driven by internal rewards and sustained by the experience of interest and enjoyment, without any obvious external rewards [55]. Additionally, we expected large correlations between the EEQ-G and reference questionnaires for game enjoyment (GExQ; H$_{A,2}$) and physical activity enjoyment (PACES; H$_{A,3}$). In line with this expectation, H$_{A,2}$ was confirmed, as we found a significant large positive correlation between the EEQ-G and the GExQ scores. H$_{A,1}$ was not confirmed, because the correlation between the EEQ-G scores and the sub-scores 'intrinsic motivation' of the BREQ was not significant. Although there was a significant positive correlation between the EEQ-G scores and the total PACES scores, H$_{A,3}$ was not confirmed, because criterion b (i.e., a validity coefficient of $|r_s| \geq 0.5$) was not fulfilled. A possible explanation for these findings is the characteristics of the collected data. The data of the sub-score 'intrinsic motivation' of the BREQ and the PACES scores were highly negatively skewed (skewness of—1.830 and– 1.240, respectively). Additionally, 21% of the participants scored four points in the sub-score 'intrinsic motivation' of the BREQ and 16% of the participants scored 80 points the PACES questionnaire. These are the highest possible scores and suggest the highest possible intrinsic motivation and physical activity enjoyment measurable in these questionnaires, respectively. According to [78, 79], ceiling effects are considered to be present if more than 15% of participants achieve the highest possible score. Based on this, it can be argued that a ceiling effect is present in the data of the sub-score 'intrinsic motivation' of the BREQ and the PACES scores. The occurrence of a ceiling effect in the sub-score 'intrinsic motivation' of the BREQ might be explained by a possible selection bias in the recruitment process (see section 'Discussion–Limitations'). Comparing our PACES data with the literature, such high PACES values are not atypical for exergame studies with HOA, where mean scores of 64% [32], 65% [35], between 80–90% [30, 31] or above 90% [29, 33] of the maximum score have been reported (compared to an average score of 96.3% of the maximum score in this study). Data of the EEQ-G was less skewed (skewness of– 0.998 and– 0.370 for the 'preferred' and 'unpreferred' condition, respectively) and none of the participants reached the maximal rating (the highest of all ratings was 93 out of 100), indicating an advantage for the EEQ-G compared to the PACES when applied for measuring exergaming enjoyment in HOA.

For discriminant validity, we formulated one hypothesis (see section 'Introduction–Objectives'). We expected no correlation between exergame enjoyment and external motivation (H$_{A,4}$), because external motivation is the least autonomous form of extrinsic motivation, is at the other end of the Self-Determination Continuum, and refers to being motivated to satisfy an external demand or a socially constructed contingency, or to avoid punishment [55]. In line with this expectation, we found no significant correlation between the EEQ-G scores and the sub-scores 'external motivation' of the BREQ. Therefore, H$_{A,4}$ was confirmed, suggesting acceptable discriminant validity of the EEQ-G. However, these results must be interpreted with caution as they have the same limitations that apply for hypotheses one and three. The data of the sub-scores 'intrinsic motivation' of the BREQ were highly positively skewed

(skewness of 2.643), and 58.1% of participants scored zero—the lowest (= best) possible score for external motivation in this questionnaire.

## Responsiveness

In this study, exergame enjoyment was rated significantly higher in the 'preferred' than the 'unpreferred' condition with a large effect size, showing that the EEQ-G was able to detect changes in the construct exergame enjoyment in relation to differing consecutive exergaming conditions. For the original English EEQ, responsiveness has not (yet) been analyzed. Therefore, our results cannot be compared in this respect. However, the EEQ ratings can be compared with two groups of study participants who played two different exergames ('Pokémon Go' and 'Just Dance Now'). A lower EEQ rating in the exergame 'Pokémon Go' (mean = 67, range: 54 to 78) compared to 'Just Dance Now' (mean = 75, range: 59 to 96) was found. This result was explained by the observation that several players complained about technical problems (e.g., Wi-Fi connection interruptions) that ultimately affected their exergame enjoyment. [36] However, the different ratings in exergame enjoyment between 'Pokémon Go' and 'Just Dance Now' could additionally be explained by other factors (e.g., participants' preferences for the games or differences in demographics variables between the two groups) and do not allow any firm conclusions about the responsiveness of the EEQ.

## Implications for research

The results of this study suggest that the construct validity of the EEQ-G needs further evaluation. In contrast to this study, the original English EEQ has been initially validated based on evaluating the agreement of the EEQ scoring with a free-form discussion about participant's exergame experience in response to open-ended questions. It has been shown that *"average (mean) of 85 percent of coded responses agreed with the subjects' statements in focus group discussions, effectively validating the EEQ"* [36]. Investigations on possible item reductions and specific evaluations on the original English EEQ including dimensionality, tests of reliability and validity are ongoing [36]. Further investigations on construct-validity, including possible revisions (e.g., item reductions), of the EEQ-G are also required.

Additionally, future research is needed to investigate the effectiveness of the EEQ to achieve its intended purposes (as defined in [36]): (1) discover determinants of exergame enjoyment, and (2) implement the EEQ to monitor exergame enjoyment with the aim to adapt the exergame training to sustain or increase exergame enjoyment over time and maximize training adherence.

## Limitations

The outcomes of this study must be interpreted with caution considering the following limitations: First, participants were recruited by contacting suitable individuals from a participant pool of our lab. This participant pool was only recently (during three studies running in our research group in the beginning of 2021) generated based on conventional recruitment procedures (such as advertising at the 'University of Third Age', senior homes and leisure-activity institutions for seniors). Although the participant pool provides sufficient diversity at ages ≥ 60 years, sex, and preferred leisure time activities, it may limit the generalizability of our results to some extent. Additionally, it may have contributed to the second limitation, that the construct validity analysis was clearly limited by the highly skewed data with ceiling/bottom effects for two of the reference questionnaires, namely the BREQ and the PACES. This might be explained by a selection bias in the recruitment process. It seems likely that mainly those who enjoyed using the system in their previous study participations (and were therefore

intrinsically motivated to use it again) or enjoy being physically active in general (69% of participants had a high level of physical activity according to the IPAQ-SF) responded to the invitation to participate in this study. Third, all participants were fully informed about the study (including the study objectives) and rank-ordered the five games right before performing the two short exergame sessions. This may have influenced their EEQ-G rating, as the participants were aware of this ranking while playing the games. To minimize bias, we asked the participants to rank-order the games according to their 'preferences' and not according to their 'perceived enjoyment'. Additionally, the two exergame sessions were completed in randomized order and the participants were not provided with any information on how these two sessions differed in content.

## Conclusion

The translated and cross-culturally adapted EEQ-G has high internal consistency and its rating is responsive to changes in exergame enjoyment in relation to different consecutive exergaming conditions. The highly skewed data with ceiling effects in some of the reference questionnaires deem the construct validity of the EEQ-G to be inconclusive and thus in need of further evaluation. Future studies should investigate the effectiveness of the EEQ to discover determinants of exergame enjoyment and implement the EEQ to monitor exergame enjoyment with the aim to adapt the exergame training to sustain or increase exergame enjoyment over time and maximize training adherence.

## Supporting information

**S1 File.**
(PDF)

## Acknowledgments

The authors would like to thank all participants in this study for their participation and valuable contribution to this project. Additionally, the authors would like to thank Lorenzo Einaudi and Kathrin Rohr for their support in data collection.

## Ethics approval

All study procedures were carried out in accordance with the Declaration of Helsinki. The study protocol was approved by the ETH Zurich Ethics Committee (EK-2021-N-135).

## Author Contributions

**Conceptualization:** Patrick Manser, Simone Huber, Julia Seinsche, Eling D. de Bruin, Eleftheria Giannouli.

**Data curation:** Patrick Manser.

**Formal analysis:** Patrick Manser.

**Investigation:** Patrick Manser.

**Methodology:** Patrick Manser, Simone Huber, Julia Seinsche, Eling D. de Bruin, Eleftheria Giannouli.

**Supervision:** Eling D. de Bruin, Eleftheria Giannouli.

**Visualization:** Patrick Manser.

**Writing – original draft:** Patrick Manser, Simone Huber, Julia Seinsche.

**Writing – review & editing:** Patrick Manser, Simone Huber, Julia Seinsche, Eling D. de Bruin, Eleftheria Giannouli.

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
