## [Decision Letter · Decision Letter 0]

17 Jan 2023

PONE-D-22-33490

Development and initial validation of the German version of the Exergame Enjoyment Questionnaire (EEQ-G)

PLOS ONE

Dear Dr. Manser,

Thank you for submitting your manuscript to PLOS ONE. After careful consideration, we have decided that your manuscript does not meet our criteria for publication and must therefore be rejected.

I am sorry that we cannot be more positive on this occasion, but hope that you appreciate the reasons for this decision.

Kind regards,

Kirubel Shiferaw, Bsc, MPH, Msc, PhD.C

Academic Editor

PLOS ONE

Additional Editor Comments:

Reviewers' comments:

Reviewer's Responses to Questions

**Comments to the Author**

1. Is the manuscript technically sound, and do the data support the conclusions?

Reviewer #1: Partly

2. Has the statistical analysis been performed appropriately and rigorously? 

Reviewer #1: No

3. Have the authors made all data underlying the findings in their manuscript fully available?

Reviewer #1: No

4. Is the manuscript presented in an intelligible fashion and written in standard English?

Reviewer #1: Yes

5. Review Comments to the Author

Reviewer #1: ID: PONE-D-22-33490

Title: Development and initial validation of the German version of the Exergame Enjoyment Questionnaire (EEQ-G).

Thank you for providing a chance to review this manuscript.

Comment: Reject.

Detailed information:

Abstract

Line 40-43, page 3: These three sentences are not well connected and the logic is very problematic. I think the "method" section needs to be rewritten to better focus the paper.

Line 32-56, page 2-4: 1) The abstract is too long. Please shorten it and keep only the most important content; 2) All statistics need to be in italics, please check for the full text.

Methods

Line 137-132, page 8-9: Was there an expert review board involved in the translation process? Was the translated version approved by the original developers of the scale? Was there a pilot test after the translation was completed?

Line 152-153, page 9: “participants who expressed interest in participating in future studies of our research group” ------ Does this have an impact on the applicability of the study findings?

Line 158, page 10: “healthy (based on self-report) older adults” ------ Each elderly person has a different standard of health and how to demonstrate that self-reported health is a uniform criterion for inclusion.

Line 210-236, page 13-15: 1) What about the psychometric properties of these scales? Have any previous studies used the German version? 2) The presentation of these scales is too confusing and I would recommend you to study the logic of other papers in TOP journals.

Line 238-245, page 15: 1) Are there any missing data in this study? What methods were used to deal with it? 2) What methods were used for quality control?

Line 252-254, page 16: I think your description of the hypothesis is unclear.

Line 273-285, page 17-18: Perhaps it would be better to put the sample size calculation in the " Participants and recruitment" section.

Results

1) The results section doesn't just describe the table all over again, and there are many phrases recurring over and over again that are very redundant; 2) The tables are irregular and unclear. Also, it should be common sense to put notes below the table and to use a three-line table to present the data;

Conclusions

Line 433-441, page 29: The results section is generally repeating what has been mentioned before, and I hope you can make a higher level summary and conclusion.

Thank you and my best,

Your reviewer

6. PLOS authors have the option to publish the peer review history of their article (what does this mean?). If published, this will include your full peer review and any attached files.

Reviewer #1: No

- - - - -

---

## [Author Response · Author response to Decision Letter 0]

23 Feb 2023

Dear Editor and Reviewer

Thank you very much for handling and reviewing our manuscript (PONE-D-22-33490) with the title: “Development and initial validation of the German version of the Exergame Enjoyment Questionnaire (EEQ-G)”.

We sincerely appreciate all your valuable comments and suggestions, which helped us to improve the manuscript's quality. However, after closely looking into the reviewer’s comments, we regret to express our concerns that some of the reviewer’s comments contained technical errors.

Our main concerns regarding the quality of the peer-review relate to the following points, as we feel that the reviewer may have missed some information in our manuscript as they stated them:

1. The reviewer states that we have not made all the data underlying the findings in his manuscript fully available. This is incorrect. As reported in lines 447 - 448 of the original submission "The datasets generated and/or analyzed during the current study are available in the Zenodo repository, https://doi.org/10.5281/zenodo.7373180."

2. Regarding the peer-review comment “Line 252-254, page 16: I think your description of the hypothesis is unclear.“: As lines 252 - 254 concern the statistical analyses, we are not sure whether the reviewer considered our definitions of the hypotheses in the section "Objectives" (lines 128 - 136 of the original submission)? If unclari- ties remain, we are open to feedback on how we can improve the clarity of our hypotheses.

3. The peer-reviewer states that the statistical analysis has not been performed appropriately and rigorously. However, no clear arguments were provided to support this evaluation. We would be very interested in the reviewer’s detailed explanation, why they rated our statistical analysis as not proper. We carefully defined all statistical analyses based on scientific literature dealing with internal consistency, construct validity, and responsiveness, and explained this with referencing the relevant scientific literature in lines 246 - 272 of the original submission.

The only peer-review comments that refer to the statistical analyses are the following: "Line 238-245, page 15: 1) Are there any missing data in this study? What methods were used to deal with it? 2) What methods were used for quality control?“

o Regarding 1): It is reported in our manuscript that there were no missing data (see figure 2 in the original submission). Additionally, it was reported that “Recruitment was stopped when complete data of the planned minimum sample size of 42 participants was available.“ (lines 282 – 283 of original submission). Therefore, there was no need for (reporting of) handling missing data.

o Regarding 2): This information is indeed missing in our original submission, so we incorporated it in the revised version of the manuscript as follows (see attachments; lines 251 - 259): "All involved study investigators were thoroughly trained for all study procedures according to Guidelines of Good Clinical Practice (GCP) and in line with detailed working instructions. The principal investigator was in charge for methodological standards and quality of data collection using data management system Castor EDC (Ciwit BV, Amsterdam, The Netherlands) [46]. Range checks for data values were pre-programmed for data entry in eCRFs. All data entries were cross-checked by a second study investigator prior to export for analysis. To minimize bias during assessment of all outcome measures, detailed working instructions were prepared that include standardized measurement procedures and standardized instructions of participants for all measurements."

We thank the reviewer for the remaining comments. Below we explain how we have either integrated these comments into our revised manuscript or how we would like to explain and resolve possible disagreements with the reviewer. In particular:

1. Regarding the peer-review comment "Line 137-132, page 8-9: Was there an expert review board involved in the translation process? Was the translated version approved by the original developers of the scale? Was there a pilot test after the translation was completed?“:

As reported in lines 138 - 139 of the original submission, "First, the original English EEQ was translated and cross-culturally adapted to German according to the “guidelines for the process of cross-cultural adaptation of self-report measures” [40]. According to these guidelines, this process includes consulting a committee of experts as well as the original developers of the questionnaire. Therefore, we (1) did obtain approval of the original developer of the questionnaire to develop a translated version of the questionnaire, and (2) the original developer approved the translated version. As this is part of the standard process defined in “guidelines for the process of cross-cultural adaptation of self-report measures” [40], we did not specifically report this in the original submission but have now clarified this in the revised version (see lines 146 - 148 of the revised manuscript: "In line with these guidelines, we obtained approval of the original developer of the questionnaire to develop a translated version of the questionnaire, and the original developer approved the translated version.“

Regarding pilot-testing: This study represents the pilot testing of the questionnaire after translation was completed and, therefore, only represents the „initial“ validation of the questionnaire (see title, and “[...] the construct validity of the EEQ-G needs further evaluation.“ (line 398 of original submission)). We also clarified this in the revised manuscript (see lines 451 - 453): "Further investigations on construct-validity, including possible revisions (e.g. item reductions), of the EEQ-G are also required"

2. Regarding the peer-reviewer comments “Line 152-153, page 9: “participants who expressed interest in participating in future studies of our research group” ------ Does this have an impact on the applicability of the study findings?“ We are aware that this point is a limitation of our study. We therefore explained how our recruitment may have affected the results in the limitation section in lines 413 - 425 of the original submission as follows: “The outcomes of this study must be interpreted with caution considering the following limitations: First, participants were recruited by contacting suitable individuals from a participant pool of our lab. This participant pool was only recently (during three studies running in our research group in the beginning of 2021) generated based on conventional recruitment procedures (such as advertising at the ‘University of Third Age’, senior homes and leisure-activity institutions for seniors). Although the participant pool provides sufficient diversity at ages ≥ 60 years, sex, and preferred leisure time activities, it may limit the generalizability of our results to some extent. Additionally, it may have contributed to the second limitation, that the construct validity analysis was clearly limited by the highly skewed data with ceiling/bottom effects for two of the reference questionnaires, namely the BREQ and the PACES. This might be explained by a selection bias in the recruitment process. It seems likely that mainly those who enjoyed using the system in their previous study participations (and were therefore intrinsically motivated to use it again) or enjoy being physically active in general (69 % of participants had a high level of physical activity according to the IPAQ-SF) responded to the invitation to participate in this study”. If further clarification or explanation should be necessary in the reviewer's view, we are open to include those.

3. Regarding the peer-review comment "Line 40-43, page 3: These three sentences are not well connected and the logic is very problematic. I think the "method" section needs to be rewritten to better focus the paper.“: We thank the reviewer for this comment and have revised the methods section of the abstract (see lines 40 - 46 of the revised manuscript).

4. Regarding the peer-review comment "Line 32-56, page 2-4: 1) The abstract is too long. Please shorten it and keep only the most important content; 2) All statistics need to be in italics, please check for the full text.“. The abstract contains 297 words and thus complies with the PLOS ONE submission guidelines. We are not aware of any requirement to italicise statistics and could not find any corresponding instructions in the PLOS ONE submission guidelines. But if necessary, we are open to change the formatting of the relevant sections in the manuscript.

5. Regarding the peer-review comment "Line 158, page 10: “healthy (based on self-report) older adults” ------ Each elderly person has a different standard of health and how to demonstrate that self-reported health is a uniform criterion for inclusion.“ We agree with the reviewer that “each elderly person has a different standard of health“. This is also in line with the WHO definition of health which states that "Health is a state of complete physical, mental and social well-being and not merely the absence of disease or infir-

mity.“ (https://www.who.int/about/governance/constitution). In that sense it is true that health cannot be defined uniformly. As such our definition of health by self-report is fully in line with the WHO definition and used in a similar way in numerous scientific reports. Moreover, the aim of our study should be considered – namely to evaluate the initial validity of a questionnaire und not to report health-related effects of an intervention. These are the reasons why we decided this definition of health was sufficient. However, in case the reviewer would like us to deviate from the WHO definition we would appreciate receiving an argumentation to do so.

6. Regarding the peer-review comment “Line 210-236, page 13-15: 1) What about the psychometric properties of these scales? Have any previous studies used the German version? 2) The presentation of these scales is too confusing and I would recommend you to study the logic of other papers in TOP journals.“o

Regarding 1): Information on the validity and reliability of each outcome measure was provided in the original manuscript, including references to the relevant publications. Additionally, we provided key references of publications that have used the respective questionnaires. In particular, we reported the following:

§ For the Behavioral Regulation in Exercise Questionnaire (BREQ): “Training motivation was assessed with the German translation [50] of the revised [51] Behavioral Regulation in Exercise Questionnaire (BREQ) [52], a widely used, valid and reliable measure of training motivation [51-55].” (lines 211 – 213 of the original submission)

§ For the Game Experience Questionnaire: “The GExQ assesses seven categories of subjective game experience (i.e. competence, immersion, flow, tension, challenge, negative affect, and positive affect) [58, 59], and has been applied in studies evaluating exergaming experience in older adults [60, 61].” (lines 224 – 226 of the original submission)

§ For the Physical Activity Enjoyment Scale: “The PACES has already been implemented for measuring exergame enjoyment in HOA [29-32, 35]. The German [63] 16-item version [64] of the PACES was used, which is valid and reliable in children, adolescents and adults [63-65].” (lines 231 – 233 of the original submission) However, the reviewer is right that we did not specifically report the psychometric properties and which other publications have been using the German versions of the reference questionnaires. Based on the peer-reviewer report, we now added this information to the revised manuscript as follows:

§ For the Behavioral Regulation in Exercise Questionnaire (BREQ): “Training motivation was assessed with the German translation [50] of the revised [51] Behavioral Regulation in Exercise Questionnaire (BREQ) [52], a widely used, valid and reliable measure of training motivation [51-55] with acceptable internal consistency (Cronbach’s α between 0.73 and 0.91 [51, 53]. The BREQ uses a Likert scale with items for motivational subscales along the Self-Determination Continuum [52, 56]. The German translation of the BREQ has been used to assess change in training motivation over time in a randomized controlled trial (RCT) investigating a life-style-integrated functional exercise intervention in older adults (n = 294). In this study, the Ger-man translation of the BREQ had a high internal consistency (Cronbach’s α of 0.84 – 0.87) [57]. Additionally, it has been used in a pilot RCT investigating a computerized cognitive training approach aiming to increase physical activity in healthy individuals [58] and a cross- sectional study testing an extension of the health action process approach by including intrinsic motivation in older adults [59].“ (lines 216 – 226 of the revised manuscript)

§ For the Game Experience Questionnaire: “The GExQ assesses seven categories of subjective game experience (i.e. competence, immersion, flow, tension, challenge, negative affect, and positive affect) with acceptable internal consistency (Cronbach’s α between 0.69 and 0.91 [38, 61], and has been applied in studies evaluating exergaming experience in older adults [62, 63].“ (lines 240 – 245 of the revised manuscript)

§ For the Physical Activity Enjoyment Scale: “The PACES has already been implemented for measuring exergame enjoyment in HOA in multiple studies [29-32, 35]. The German [65] 16-item version [66] of the PACES was used, which is valid and reliable in children, adolescents and adults [65-67], with high internal consistency (Cronbach’s α between 0.89 to 0.94) [65, 67]. The German PACES has been used in numerous studies, as for example in RCTs investigating the effects of different types of motor-cognitive training on cognitive performance in older adults [68, 69], or to assess psychological and physio-logical responses to exergaming [70].” (lines 248 – 254 of the revised manuscript)

o Regarding 2): Unfortunately, we do not understand the comment of the peer-reviewer and would therefore like to ask for an additional explanation on suggested changes in the method section for us to be able to integrate the feedback of the reviewer.

6. Regarding the peer-review comment "Line 273-285, page 17-18: Perhaps it would be better to put the sample size calculation in the „ Participants and recruitment" section.“ The sample size calculations are based on the specific statistical methods used in the study, so for a clear text structure, they should be reported after these statistical methods are defined.

7. Regarding the peer-review comment "1) The results section doesn't just describe the table all over again, and there are many phrases recurring over and over again that are very redundant; 2) The tables are irregular and unclear. Also, it should be common sense to put notes below the table and to use a three-line table to present the data;“ Regarding 1): We double-checked whether there are any repetitions of information that are given both in the tables/figures and in the main text in the results section and did not find any redundancies in our original submission. We would be happy to shorten/adjust if you point us out to specific sections. Regarding 2): We indeed notice that we inserted our table captions below the table title and before the actual table, which does not meet the PLOS ONE submission guidelines. Therefore, we corrected this in the revised manuscript. Regarding the peer-reviewer's feedback that the tables are "irregular and unclear" we would kindly like to ask you to provide further explanations on suggested changes in the tables.

8. Regarding the peer-review comment "Line 433-441, page 29: The results section is generally repeating what has been mentioned before, and I hope you can make a higher level summary and conclusion.“ We do not agree with this feedback of the peer-reviewer. Our conclusions are supported by the data presented in our manuscript and summarize our main arguments and findings as well as the key takeaways from our paper, which is in line with the PLOS ONE submission guidelines.

We kindly ask you to consider the following clarifications and reconsider your decision and / or invite additional peer-reviewers to evaluate our work.

Many thanks for your time and consideration. 

Kind regards (on behalf of all my co-authors),

Patrick Manser | Doctoral Student

ETH Zurich | Department of Health Sciences and Technology Institute of Human Movement Sciences and Sport

Motor Control and Learning Group

HCP H24.3

Leopold-Ruzicka-Weg 4 | 8093 Zurich | Switzerland

E-Mail: patrick.manser@hest.ethz.ch ǀ https://mcl.ethz.ch

Phone: +41 79 519 96 46

---

## [Editor Report · Decision Letter 1]

19 May 2023

Development and Initial Validation of the German Version of the Exergame Enjoyment Questionnaire (EEQ-G)

PONE-D-22-33490R1

Dear Dr. Manser,

We’re pleased to inform you that your manuscript has been judged scientifically suitable for publication and will be formally accepted for publication once it meets all outstanding technical requirements.

Kind regards,

Mariam Ahmad Abu Alim, PhD

Academic Editor

PLOS ONE
---

## [Editor Report · Acceptance letter]

25 May 2023

PONE-D-22-33490R1 

Development and Initial Validation of the German Version of the Exergame Enjoyment Questionnaire (EEQ-G) 

Dear Dr. Manser:

I'm pleased to inform you that your manuscript has been deemed suitable for publication in PLOS ONE. Congratulations! Your manuscript is now with our production department. 

Kind regards, 

on behalf of

Professor Mariam Ahmad Abu Alim 

Academic Editor

PLOS ONE